# Bullying Victimization Moderates the Association between Social Skills and Self-Esteem among Adolescents: A Cross-Sectional Study in International Schools

**DOI:** 10.3390/children9111606

**Published:** 2022-10-22

**Authors:** Yunru Dou, Tinakon Wongpakaran, Nahathai Wongpakaran, Ronald O’Donnell, Saifon Bunyachatakul, Pichaya Pojanapotha

**Affiliations:** 1Master of Science (Mental Health), Graduate School, Chiang Mai University, Chiang Mai 50200, Thailand; 2Department of Psychiatry, Faculty of Medicine, Chiang Mai University, Chiang Mai 50200, Thailand; 3Behavioral Health, College of Health Solutions, Arizona State University, 500 N. 3rd St, Phoenix, AZ 85004, USA; 4Department of Occupational Therapy Faculty of Associated Medical Sciences, Chiang Mai University, Chiang Mai 50200, Thailand

**Keywords:** victim, bully, international school, social skills, self-esteem, moderation model

## Abstract

**Background.** Bullying is a major school problem. Victims of bullying often experience low self-esteem, whereas social skills are positively associated with the level of self-esteem. This research examined whether the victim’s condition impacted their social skills and self-esteem. **Methods.** International school students in Thailand aged 13 to 18 years old completed the Olweus bullying questionnaire, social capital questionnaire (SC), social skills questionnaire (SS), adolescent discrimination index (ADDI), and the Rosenberg self-esteem scale (RSES). Moderation analyses and visual presentations were carried out using IBM SPSS ver. 22 and PROCESS, ver. 4.0. **Results.** A total of 102 students participated (63% female). The mean age of the participants was 16.57 (SD = 1.42). The number of victims was 16 (15.7%), the mean (SD) for the SC, SS, ADDI, and RSES was 7.82 (2.37), 44.45 (9.40), 12.33 (9.82), and 27.85 (5.31), respectively. As predicted, those with high social skills reported greater self-esteem when they had never been bullied. The moderation effect was significant: B = 0.458, standard error = 0.203, 95% CI = −0.836 to −0.054. Additionally, the ADDI and SC were found to predict self-esteem.** Conclusions.** The significant moderation effect suggests the importance of identifying the victim’s condition when the association between social skills and self-esteem is not observed (as expected) among school adolescents. A longitudinal study to confirm the causal relationship should be encouraged. Further research on providing appropriate interventions along with social skill training for the victim group is warranted.

## 1. Introduction

### 1.1. Bullying and Prevalence

Bullying is an important problem in schools [1]. It constitutes a form of violence and is encouraged by aggressive, repetitive behaviors and imbalanced power relations [2]. The many types of school bullying include physical bullying, such as beating, physical injury, and sexual abuse [3], and verbal bullying, such as name-calling and taunting, or indirect bullying, such as rumor-mongering and online bullying [2].

The prevalence of bullying varies and may be inconsistent, depending on the different dimensions of bullying or changes in demographic characteristics [4]. In Thailand, a survey of Thai global school student showed that 27.8% had experienced bullying [5]. Bullying is a major problem and can be found in schools regardless of being public or private, elementary or secondary, and urban or rural [6].

The school system in Thailand has developed locally with a substantial focus on Thai culture and values. Thailand is a collectivistic culture [7], and Thais tend to be more altruistic and helpful than people in individualistic cultures, with these characteristics being attributable to Theravada Buddhism [8]. Seniority and social hierarchy may play a role concerning the school environment in public schools in Thailand compared to those in Western cultures [9]. However, no evidence has been reported regarding how culture impacts on bullying.

In Thailand, schools using a Western education model were only just established in the early 19th century. Part of the reason is that Thailand is one of the few countries in the world that Western powers have never colonized. In recent years, international education in international schools has become widespread and has risen rapidly [10]. However, the system is deemed for expatriates and high-income families, whereas a public school education in Thailand (until Grade 9) is free of charge. It has been generally accepted that international schools in Thailand abide by international standards offering high-quality campuses, modern facilities, a plethora of learning resources, and a variety of extracurricular activities. Because of this, bullying is expected to be nonexistent.

However, bullying is a multifaceted issue, which is best understood in the societal context in which it occurs [11,12], including different ethnic and cultural backgrounds. Even though such differences may be an advantageous side to learning, they can also constitute a downside as it is open to racial discrimination, leading to bullying [13]. Other factors include socialization—how well individuals socialize with others and the number of friends and students socializing with friends who bully. Taken altogether, this raises questions regarding bullying in this setting. Although few reports have been published on bullying in international schools, it has been hypothesized that bullying might exist.

### 1.2. Bullying and Associated Factors

Many factors contribute to bullying and being bullied, such as physical appearance, weight, race or ethnicity, sex, sexual orientation, religion, and disability [14,15,16]. Social factors, e.g., network size and the quality of having a relationship with friends, are also important according to the social capital theory [17].

Social capital theory denotes the benefits obtained from social relations. According to this theory, individuals invest in social relations to obtain the resources implanted in these relations. This theory is also applicable in schools. Adolescents who are bullied have very low social status because they lack friends, resulting in minimal social capital. Bullies tend to obtain social capital more easily and use bullying tactics in the form of social status as a means of accumulating social capital [17].

Racial and cultural discrimination, especially in international schools, where students come from different cultural backgrounds, could lead to racial and cultural bullying through a variety of overt forms of aggression [18]. African American adolescents have higher rates of bullying and victimization compared with other adolescent populations [19], whereas some studies revealed that Asian youths were more likely to be victimized compared with Hispanic youths [20]. Moreover, the relationship between discrimination and self-esteem has been documented among some racial and ethnic minority populations [21].

### 1.3. Impact of Bullying

Bullying is also closely related to adolescent depression, anxiety, insomnia, inattention, and other mental problems [6]. Another cohort study of 1790 participants showed that bullying in adolescence was related to depression in adulthood, with those who had been bullied at the ages between 15 and 18 most likely to have depressive symptoms at the age of 28. A relationship was observed between bullying and depressive symptoms later in life over time [22].

Some children can self-adjust. Some require intervention to improve or to prevent complications. The results of bullying, physical and emotional, have a huge impact on childrens’ self-esteem [6].

### 1.4. Bullying and Self-Esteem

Self-esteem, an important form of happiness, is a person’s assessment of their own worth or importance. For teenagers, good self-esteem helps them believe in themselves and is important for building good social relationships. Many studies have demonstrated the association between victimization and low self-esteem. A study in municipal schools found a higher incidence of bullying in 53.7% of teens with low self-esteem and a lower average self-esteem score for both the bully and the victim [6]. A meta-analytic study found that high self-esteem was associated with reduced victimization from bullying and reduced criminal behavior. Another study found that low self-esteem is also associated with self-victimization and racial–cultural victimization [23]. In addition to traditional bullying, cyberbullying victimization contributes to predicting low self-esteem and psychological distress over and above other experiences of bullying in schools or other settings [24]. This finding is consistent with that of a study in China that discovered that adolescents with psychosocial problems were more likely to experience bullying victimization. This study also noted that stress was a specific predictor of traditional bullying victimization, whereas self-esteem, social anxiety, and loneliness were specific predictors of cyberbullying victimization [25].

### 1.5. What Influences Victimization and Self-Esteem?

Although many factors were found to be related to either victimization or self-esteem, studies directly addressing factors moderating the relationship between bullying victimization and self-esteem remain limited. One study revealed that rumination, including a focus on the past, moderated the effect of cyberbullying victimization on psychological well-being [26]. In contrast, another study showed the reciprocal relationships between adolescents’ school victimization and self-esteem in that both variables predicted each other over different time frames. In contrast, teacher support was the mediator between these two variables [27].

### 1.6. Social Skills, Self-Esteem and Bullying

Social skills have been well-documented to be related to increased self-esteem [28,29], and social skills training has significantly enhanced self-esteem [30].

Skill building increases the self-esteem of the bullied students (victims) [31,32,33]. Social skills, especially those improving conflict resolution, can protect against the possibility of being bullied and bullying behaviors [23]. Social skill interventions had positive effects on increased self-esteem and reduced bullying behavior [34]. Social skills are evidenced to play a mediating role in the relationship between bullying behaviors and victimization bullying behaviors [35]. Even though both social skills and victimization have effects on self-esteem, it remains unclear whether the relationship between social skills and self-esteem depends on the condition of being bullied or not. To put it in other words, whether there is an interaction (or moderation) effect between social skills and victimization on self-esteem. More importantly, such a relationship in a setting like an international school, where the environment and socioeconomic status is unique and different from public school in Thailand, remains unknown.

As it is already known, the effect of bullying victimization on psychological distress is enormous, to the extent that some might experience suicide. In addition, low self-esteem, fear, distress, and depression can commonly occur. We propose that among adolescents who have never been bullied, social skills might directly affect their self-esteem level, as acquiring more skills should provide individuals with confidence and a sense of accomplishment. On the contrary, those experiencing being bullied might not gain the same benefit as the nonvictim because their sense of self-worth might be compromised. Therefore, the relationship between social skills and self-esteem in the latter group might not have a significant effect. This study aimed to explore the moderation effect of the victim’s condition and social skills on self-esteem. The authors hypothesized that the moderating effect of victimization on the relationship between social skills and self-esteem would exist among adolescents. The authors’ other hypotheses included the positive correlation between social capital and self-esteem and a negative correlation between discrimination and self-esteem.

## 2. Materials and Methods

This study employed a cross-sectional and correlational design conducted among students aged 13 to 18 in international schools in Thailand from 2021 to 2022.

### 2.1. Participants and Procedures

The participants were male or female students aged 13 to 18 attending international schools in Thailand. Inclusion criteria comprised students who could understand, read, and write English fluently. The exclusion criterion consisted of students with visual impairment. The correlation between bullying victims and self-esteem was approximately 0.3 [36] based on a meta-analysis of 936 studies investigating the relationship between self-esteem and bullying behavior (i.e., perpetration and peer victimization). We set the type I error (alpha) at 0.05 and the type II error (Beta, 1-power) at 0.20; therefore, the minimum sample size was 84. Thailand hosts 119 international schools [37]. Due to the COVID-19 situation, online survey and convenience sampling were employed using the snowball method. The participants accessed the online survey using a URL link or QR code. The two forms included one for participants and another for the parent or guardian. For the participants, the online content included a consent form and questionnaires, whereas, for the parent or guardian, it comprised only a consent form. Both completed the documents independently.

According to EC, the investigators and research assistants approached the participants. The title of the project was destigmatized using a more neutral statement as per “Mental health and relationship with friends in an international school”, and both participants and parents (or guardians) provided written consent to participate in the study. To address the risk (albeit a minimal one), it was stated in the consent sheet that if the participant felt uncomfortable regarding the questionnaire, he/she was able to contact the principal investigator for consultation and advice (24 h).

Researchers sent the QR codes for the research materials to volunteers, and the guardians of the volunteers completed the PIS online. A compensation of 100 THB (approximately 3 USD) was offered for the participants.

### 2.2. Measurement

In addition to the questionnaires regarding sociodemographic data, the following measurements were provided for the participants (see the Appendix A).

#### 2.2.1. Olweus Bully/Victim Questionnaire (Olweus, 1996)

This tool looks at seven types of bullying, including bullying, verbal abuse, ostracizing (expelling from a group), physical abuse, spreading false rumors, theft/damage of personal items, threats/coercion, and race-related harassment. Altogether, 40 questions comprised the information collected from the students, teachers, and parents. The respondents were divided into nonbully nonvictim, victim bully, and bully victim groups. Researchers have found good internal consistency for the OBVQ (Cronbach’s alpha ranged from 0.8 to 0.9) [38]. The present sample showed a Cronbach’s alpha of 0.91.

#### 2.2.2. Rosenberg Self-Esteem Scale (RSES)

The RSES comprised a 10-item tool evaluating self-worth by rating positive and negative feelings about oneself. Each item employed a four-point Likert scale format, ranging from strongly agree to strongly disagree. The higher the score that the participants reported, the higher the level of self-esteem [39]. The internal consistency of the RSES was good (Cronbach’s alpha was 0.86) [40], and the present sample showed a Cronbach’s alpha of 0.86.

#### 2.2.3. Social skills Questionnaire (SSQ)

The SSQ aimed to investigate teenagers’ ability to interact with others, their ability to control their emotions, and their social enthusiasm. The tool is divided into three versions that are assessed by teenagers (the version we used in this research), parents, and teachers. It includes 30 items. The scoring options range from 0 (incorrect), 1 (sometimes true) and 2 (mostly true). Total scores ranged from 0 to 60. High scores were associated with higher levels of social skills. An assessment of the social skills of autistic toddlers showed a Cronbach’s alpha of 0.96 [41]. The present sample showed a Cronbach’s alpha of 0.96.

#### 2.2.4. Adolescent Discrimination Distress Index (ADDI)

The ADDI measures adolescents’ struggles with discrimination and the stress of dealing with racial discrimination. The ADDI consists of 15 items divided into three subscales: institutional discrimination (six items), educational discrimination (four items), and peer discrimination (five items). Each question was given a five-point scoring scale, with items adding up to an overall score, with higher scores indicating greater pain from discrimination [42]. The present sample showed a Cronbach’s alpha of 0.80.

### 2.3. Statistical Analysis

Descriptive analysis was used to examine demographic data, e.g., age, sex, education, and type and prevalence of bullying among the participants, e.g., percentages, mean, standard deviation, and scores of the different variables.

Student’s t-test was applied in assessing the difference between continuous and noncontinuous variables, e.g., mean scores of RSES between groups of students who have been bullied compared with those who have not. The association among the variables was investigated using Pearson’s correlation for continuous variables, e.g., the total score of bullying and self-esteem. Point biserial coefficients were used to study the correlation between the continuous variables and nominal variables, e.g., bullying scores and sex. Multiple regression analysis and moderation analysis were used to identify significant predictors for the RSES score.

When analyzing the moderation models, we began by examining the magnitude of the relationships among the victim’s conditions, self-esteem, and social skill scores using zero-order correlations. For moderation analysis, we used Model 1 based on Hayes [43]; the antecedent variable (X) was social skills, the outcome variable (Y) was self-esteem, and the victim’s condition was considered a moderator (W) (Figure 1). Significant interaction (social skills scores by victim condition) was examined using visualizing predicted values of RSES within the presence or absence of victim conditions [43]. We used resampling or bootstrapping, and the product of coefficient strategies, as suggested by Preacher and Hayes, when conducting moderation analyses [43,44].

We used PROCESS, Version 4.1, an add-on statistical analysis for SPSS created by Hayes [45]. For interpretation, PROCESS provides standard errors, *p*-values, and confidence intervals for the direct effect coefficients and bootstrap confidence intervals. Confidence intervals that did not straddle zero indicated statistical significance. For all the analyses, the level of significance was set at *p* < 0.05. All statistical analyses were performed using the program IBM SPSS, Version 22.0.

## 3. Results

Among the 102 participants, 63.7% were females. The average age of the participants was 16.55 years (SD = 1.39). Most of the sample were in grade 12 and attended day school. Over half reported that their academic performance was good to excellent. Over 90% reported no physical or mental illness, but about 20% experienced alcohol abuse. Most of the participants had never been bullied (84.3%) (Table 1).

Regarding the relationship between sex and bullying, 37 males (83.8%) and 65 females (90.8%) had never bullied others; a total of 12 participants were involved with bullying. The Pearson chi-square test indicated no significant relationship between sex and frequency of bullying, Pearson’s χ^2^(1) = 1.108, *p* = 0.292. For victim experience, 31 males (83.8%) and 55 females (84.6%) had never been bullied; a total of 16 participants reported being victims. The Pearson chi-square test indicated no significant relationship between sex and frequency of being a victim, Pearson’s χ^2^(1) = 0.012, *p* = 0.912. In the bystander group, 21 males (56.8%) and 38 females (58.5%) had never been bystanders. A total of 43 participants reported being bystanders. The Pearson chi-square test indicated no significant relationship between sex and frequency of being a bystander, Pearson’s χ^2^(1) = 0.028, *p* = 0.867.

Table 2 shows the mean and standard deviation between victims and nonvictims. A significant difference for the self-esteem scores between these two conditions was observed (*t* = 2.779, *df* = 100, *p* = 0.007), but not for social skills scores (*t* = 1.276, *df* = 100, *p* = 0.205).

Table 3 shows the mean and standard deviation of each variable. Victim status was positively related to self-esteem scores (*p* < 0.01). In contrast, self-esteem scores were positively related to social capital and social skills but negatively related to discrimination index scores (*p* < 0.01). Being male was associated with social capital (*p* < 0.05) but not with other variables.

We tested each regression model and found the significant predictor of the interaction effect for victims and social skills scores (B = −0.458, *p* < 0.05) when sex, age, discrimination, and social capital scores were controlled for. This interaction signifies that the social scores and self-esteem relationship depend on the existence of victim status. For example, model 3 indicated that the variances of self-esteem increased from 23.8% to 33.2% when the interaction term was added to the model, denoting the effect victim status has on this relationship.

Notably, discrimination and social capital scores were significant predictors of self-esteem scores (Table 4).

In Figure 2, the slope of the victim group was nonsignificant (B = −0.310, SE = 0.176, *df* = 91, *p* = 0.082), whereas the slope of the nonvictim group was significant (B = 0.191, SE = 0.055, *df* = 91, *p* < 0.001). The interaction terms were significant (B = −0.501, SE = 0.186, *t* = −2.692, *p* = 0.008 (95%CI = −0.870, −0.132).

## 4. Discussion

This study’s main hypothesis was to explore the bullying victim’s condition regarding the relationship between social skills and self-esteem. The findings confirmed that the association between the level of social skills and the level of self-esteem depends on the bullying victim’s condition. For example, the relationship was expectedly positive among adolescents who had never been bullied. In contrast, the relationship between social skills and self-esteem was insignificant among students who had been bullied. Bullying victims’ conditions appear to be the third variable influencing the relationship because moderating effects between bullying victims’ conditions and social skills are significant. These results highlight the effect of the victimization experience on self-esteem through, in this case, social skills. The psychological effect of the victimization experience may persist even though the penetrations have ended [11]. In fact, social skills are merely one example involved in bullying victimization and self-esteem. Other promoting behaviors or skills might be observed in self-esteem along the same line as social skills. Therefore, it would be essential not to allow bullying victimization to go “undetected”.

In general, social skills provide confidence for those having mastered them [46,47]. However, research findings concerning the relationship between social skills and self-esteem among adolescents were seemingly inconsistent. One study revealed significant effects of social skills training programs on raising self-esteem and decreasing physical aggression [48]. Another study showed that social skills training did not increase self-esteem among these male adolescents [49]. Notably, both studies did not obtain bullying data in their studies. The authors assumed that those adolescents in the latter study might have been bullied because a positive association between social skills training and self-esteem was not illustrated as expected. Many skills-based bullying interventions aim to improve social skills among victims of bullying; this will not stop bullying from happening but could improve resilience. However, if the trauma from bullying is deep, it may not be recovered easily and may require additional and special intervention.

The authors hypothesized that being a bully victim may lead to psychological trauma, and individual resilience might play a role in helping a bully victim get back on track [50]. Another explanation concerns post-traumatic growth that might not be sufficiently achieved. For example, psychological trauma due to bullies may persist and interfere with the growth of a sense of self even though they receive some intervention or training in social skills [51].

Interestingly, some investigators found contrasting results. Students with low self-esteem and a lack of confidence in their social and communication skills were more likely to be victims of bullying [52]. In such cases, self-esteem tends to mediate the relationship between social skills and being a bullying victim. Quite possibly, a situation could occur either way or even involve a vicious cycle. Conversely, adolescents with low self-esteem can be potential victims because they lack the social skills to defend themselves. However, the present study suggests that identifying the bullying victims’ status is crucial so that additional or special interventions may be provided along with social skills training. It would be likely that these adolescents, prone to be bullying victims, might experience aforehand psychological trauma or personality problems, requiring recovery as posttraumatic growth before acquiring further new skills [51].

The second hypothesis confirmed the existing relationship between social capital and self-esteem. Consistent with the related studies, social capital, defined by the number of friends, trusting friends, and the extent to which an individual feels autonomy and independence, did predict the level of self-esteem but was unrelated to the victim or nonvictim status. The related evidence has shown that such results are inconsistent [53,54]. The authors ascribed these findings to the relatively high-level socioeconomic status and low prevalence of bullying among the participants in such types of schools.

The third hypothesis confirmed the existing relationship between discrimination and self-esteem. In line with many studies, discrimination was found to be a significant negative predictor of self-esteem [55]. Researchers have also found an association between racial discrimination and other outcomes among children and adolescents, e.g., well-being or health [56,57] and as a risk factor for depressive symptoms and substance abuse [58]. This study explicitly noted racial discrimination in an international school. Even though racial discrimination was unrelated to victimization, it remained significantly associated with self-esteem. Further research regarding strategies to reduce the strength of the relationship between discrimination and low self-esteem should be warranted.

### 4.1. Implications and Future Recommendations

The finding suggests the importance of identifying a bullying target’s condition for self-esteem through social skills. Self-esteem associations were not observed as expected among school adolescents. Because self-esteem links to other mental health issues, especially depression [59], screening for bullying targets should be prioritized. The matters of victimization experience are essential and can be underdetected. Implementing screening programs can be carried out using a variety of methods and levels. For example, teachers or school counselors can adopt a tactful approach to elicit such information from students, especially those with low self-esteem or who seem not to benefit from self-promoting activities. In contrast, school administrators may proactively devise an intermittent friendly screening tool to derive the student’s victimization experience as well as racial discrimination without questioning its existence in such a setting. The individual characteristics of students, family members, teachers, and schools need to be targeted in bullying prevention programs to reduce bullying and victimization in schools. Future research may focus on identifying those variables moderating the bullying target’s status, e.g., perceived social support [60,61]. Additionally, it is also recommended to explore the issue of cyberbullying and compare private and public school experiences while investigating the “nature of bullying” in Thailand.

### 4.2. Strength and Limitations

Although many studies have been conducted regarding the association between social skills, the bullying target’s condition, and self-esteem, this study comprised one of the first to demonstrate the interaction effect of being a bullying target and social skills concerning their level of self-esteem. This positively contributed to our attempts to break through the bullying target’s social skills and self-esteem, thereby intervening and reducing the incidence and adverse effects of bullying. This study encountered limitations. First, the finding cannot represent the international school student due to the small number of subjects and the nonrandom sampling method. Second, as almost all international schools are private and for-profit organizations, reputation became an issue; therefore, recruiting participants was challenging for the researcher. Third, the varying school factors were not accounted for in the analysis, and, finally, the cross-sectional design could not conclude or infer a causal relationship.

## 5. Conclusions

This study demonstrated the significant moderation effect of a bullying victim’s condition on the relationship between social skills and self-esteem, denoting that the relationship between social skills and self-esteem depended on the status of bullying victimization. The study practically highlighted the importance of identifying a bullying victim’s condition when social skills and self-esteem associations among school adolescents were not observed as expected. As anticipated, a significant relationship was observed between social capital, discrimination, and self-esteem. However, the small sample size and convenience sample prevent it from constituting a representative sample of international students in Thailand. In addition, the cross-sectional design limits the determination of a causal relationship among the variables. Future studies with a more considerable sample size with random sampling should be encouraged. A longitudinal study is required to confirm a causal relationship. Additionally, it is recommended to explore the issue of cyberbullying and compare private and public school experiences. Moreover, research on providing appropriate interventions and social skills training in the bullying victim group is warranted.

## Figures and Tables

**Figure 1 children-09-01606-f001:**
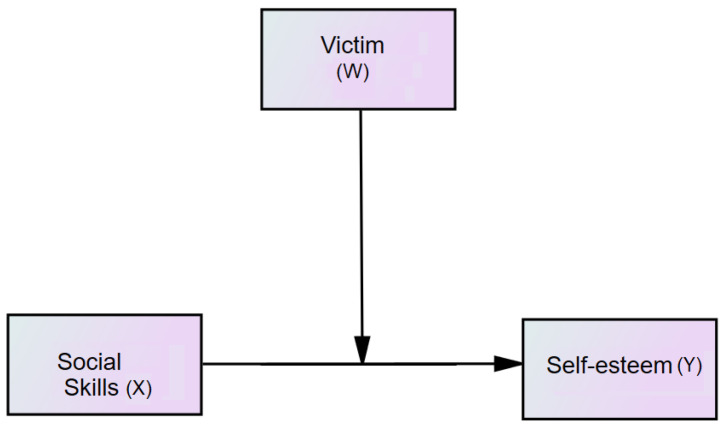
Moderation model of social skills.

**Figure 2 children-09-01606-f002:**
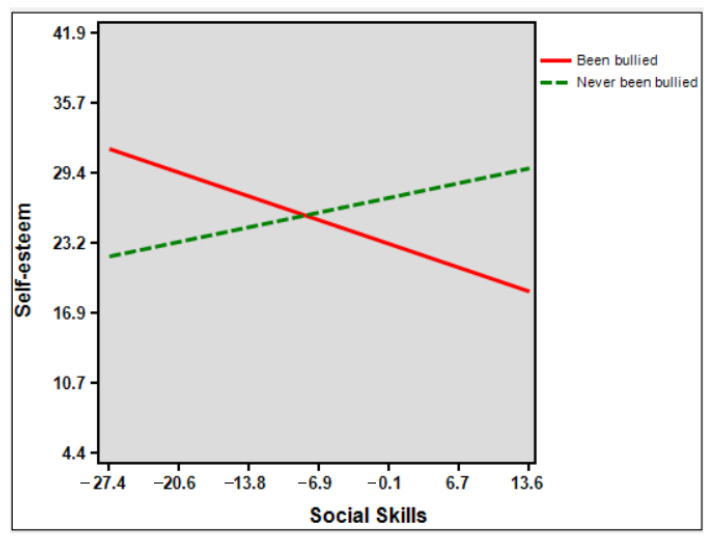
Visual representation of the moderated effect of victim conditions (X) on self-esteem scores (Y) by social skills (W).

**Table 1 children-09-01606-t001:** Sociodemographic data of the participants.

Variable	N (%) or Mean (SD)
Age	16.55 (1.39)
Grade level	
- 7	6 (5.9)
- 8	4 (3.9)
- 9	5 (4.9)
- 10	17 (16.7)
- 11	29 (28.4)
- 12	41 (40.2)
Sex	
- Male	37 (37.0)
- Female	63 (63.0)
School type	
- Day	99 (98.0)
- Boarding	2 (2.0)
Physical illness	
- No	97 (98.0)
- Yes	2 (2.0)
Reported mental problems	
- No	94 (93.1)
- Yes	7 (6.9)
Substance abuse	
- None	67 (66.3)
- alcohol	20 (19.8)
- Others (cigarette, cannabis)	14 (14.0)
Academic performance	
- Excellent	12 (11.9)
- Good	53 (52.5)
- Fair	33 (32.7)
- Poor	2 (2.0)
- Other	1 (1.0)
Had been bullied in 2 months	16 (15.7)
- Only once or twice	12 (11.8)
- 2 or 3 times a month	1 (1.0)
- Once a week	1 (1.0)
- Several times a week	2 (2.0)

*n* = 102, SD = standard deviation.

**Table 2 children-09-01606-t002:** Mean and standard deviation for self-esteem and social skills scores according to victim conditions.

Victim Condition		Self-Esteem Scores	Social Skills Scores
Victim	Mean (SD)	24.56 (4.73)	41.70 (7.14)
	n	16	16
Nonvictim	Mean (SD)	28.45 (5.21)	44.95 (9.71)
	n	86	86
Total	Mean (SD)	27.84 (5.31)	44.44 (9.39)
	n	102	102

SD = standard deviation.

**Table 3 children-09-01606-t003:** Zero order correlations between variables.

	Mean (SD) or n (%)	1	2	3	4	5	6	7
Age	160.57 (10.42)	-	0.217 *	0.246 *	0.146	0.047	0.137	0.086
2. Sex, female	65 (630.7)		-	0.134	0.108	0.011	0.100	−0.033
3. Social Capital	70.82 (20.37)			-	0.201 *	0.162	−0.021	0.280 **
4. Social Skills	440.45 (90.40)				-	0.127	−0.017	0.314 **
5. Victim	16 (150.7)					-	−0.177	0.268 **
6. Discrimination index	120.33 (90.82)						-	−0.308 **
7. Self-esteem scale	270.85 (50.31)							-

* *p* < 0.05, ** *p* < 0.01, SD = standard deviation.

**Table 4 children-09-01606-t004:** Various models estimating self-esteem scores.

Model		B	SE	*t*	*p*-Value	LLCI	ULCI
1	Constant	13.802	5.451	2.532	0.011	3.634	25.068
Adj R-sq = 0.224	Social skills (X)	0.138	0.055	2.509	0.014	0.036	0.255
2	Constant	14.770	5.553	2.660	0.008	4.022	26.075
Adj R-sq = 0.238	Social skills (X)	0.131	0.057	2.298	0.022	0.026	0.250
	victim (W)	−2.201	1.440	1.528	0.124	−5.029	0.652
3	Constant	19.122	5.805	3.294	0.001	7.595	30.648
Adj R-sq = 0.332	Social skills (W)	0.172	0.571	3.019	0.003	0.059	0.286
	victim (X)	−3.433	1.561	−2.199	0.030	−6.532	−0.333
	X W	−0.458	0.204	−2.250	0.026	−0.863	−0.053
	Sex (Male)	−1.371	0.930	−1.475	0.144	−3.217	0.475
	Age	0.517	0.324	1.596	0.114	−0.1263	1.161
	Discrimination	−0.144	0.054	−2.657	0.009	−0.252	−0.036
	Social Capital	0.404	0.188	2.154	0.034	0.032	0.777

Adj R-sq = adjusted r square, SE = standard error, B = unstandardized coefficient, X = predictor, W = moderator, LLCI = lower level of confidence interval, ULCI = upper level of confidence interval.

## Data Availability

The datasets used and/or analyzed during the current study are available from the corresponding author upon reasonable request.

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
