# Peer review of "Bullying Victimization Moderates the Association between Social Skills and Self-Esteem among Adolescents: A Cross-Sectional Study in International Schools"

_children, 2022, doi:10.3390/children9111606_

Round 1

Reviewer 1 Report

This article discusses the role of victimisation as a potential moderator of the relationship between social skills and self-esteem in international schools. 

The introduction is relatively sound, referenced some key studies in the field. The introduction starts with "bullying" then moves to discussing self-esteem and social skills. International schools is the assumed originality of this study, but i am not convinced that international schools are much from any other school, pressure on children/teens comes from parents most of the time, regardless of which school they participate in. Perhaps this could be argued better!  The relationships between bullying, social skills and self-esteem are well documented in research perhaps under different terms. There are many skills-based bullying interventions that aim to improve social skills among victims of bullying; that will not stop bullying from happening, but could improve "resilience". 

Cyberbullying is currently more prevalent compared to "traditional" bullying; i don't think that was reflected enough in the introduction. I also wonder why this was not considered in the hypothesis! 

Methods: Generally good, although i ethical consideration are not elaborated enough. What if participants disclosed worrying issues in relevance to the questionnaire? was there a method to deal with such stresses?  The sample size is a challenge here, 102 participants, is considered a small sample size when comparing your research outcomes against other publications. 

Results: Descriptive stats show that there are only 16 victims of bullying compared to 86 non-victims. Hence, it is hard to generalise the findings in this study. Although the data and stats are reported correctly, i am a little worried about the generalisability of this study.

Discussion is rather brief, and lacks theoretical explanations to the main research outcomes. Implications and future recommendations could be elaborated. 

Author Response

Reviewer 1

This article discusses the role of victimisation as a potential moderator of the relationship between social skills and self-esteem in international schools. 

The introduction is relatively sound, referenced some key studies in the field. The introduction starts with "bullying" then moves to discussing self-esteem and social skills. International schools is the assumed originality of this study, but i am not convinced that international schools are much from any other school, pressure on children/teens comes from parents most of the time, regardless of which school they participate in. Perhaps this could be argued better!  The relationships between bullying, social skills and self-esteem are well documented in research perhaps under different terms. There are many skills-based bullying interventions that aim to improve social skills among victims of bullying; that will not stop bullying from happening, but could improve "resilience". 

Response: Thank you for pointing these points out. We have revised by adding more on the uniqueness of the international in Thailand which might be different from international school in western countries. Also, the point that social skills could improve resilience is good, and we have added it in our discussion.

It now reads,

The school system in Thailand has developed locally with a substantial focus on Thai culture and values. Schools with a Western education model were just infused in the early 19th century. Part of that is because Thailand is one of the few countries in the world that Western powers have never colonized. In recent years, international education in international schools has been widespread and has risen rapidly[7]. However, it is deemed for expatriates and high-income families, whereas a public-school education in Thailand until Grade 9 is free of charge. In addition, the students in international schools come from different ethnic and cultural backgrounds. Although low socioeconomic status may be significantly associated with bullying, it does not mean that bullying never exists among adolescents with high socioeconomic status, as in international schools[8-10]. However, reports regarding bullying in international schools, either in Thai or abroad, remain limited.

Many skills-based bullying interventions aim to improve social skills among victims of bullying; that will not stop bullying from happening but could improve resilience. 

Cyberbullying is currently more prevalent compared to "traditional" bullying; i don't think that was reflected enough in the introduction. I also wonder why this was not considered in the hypothesis! 

Response. Thank you for this suggestion. We have covered both traditional and cyberbullying, even though not extensively because our present study focuses mainly on the association between social skills and self-esteem.

Methods: Generally good, although i ethical consideration are not elaborated enough. What if participants disclosed worrying issues in relevance to the questionnaire? was there a method to deal with such stresses?  The sample size is a challenge here, 102 participants, is considered a small sample size when comparing your research outcomes against other publications. 

Response. Thank you for pointing this out. We have added more information on how the participants get help when needed. These statements are added.

To address the risk, albeit minimal, it was stated in the consent sheet that if the participant felt uncomfortable in relevance to the questionnaire, he/she was able to contact the principal investigator for consultation and advice (24 hours).

 Regarding the sample size, we agree that it was small, which would affect the population's prevalence. We have put that in the limitation part.

Results: Descriptive stats show that there are only 16 victims of bullying compared to 86 non-victims. Hence, it is hard to generalise the findings in this study. Although the data and stats are reported correctly, i am a little worried about the generalisability of this study.

Response. We agree with generalizability, especially regarding the prevalence and have stated that in the limitation. That is why we did not focus on the distribution related to the type of bullying, instead paying attention to the association between variables (including the moderation effect), which we believe, with this sample size, still has sufficient statistical power for such testing.

Discussion is rather brief, and lacks theoretical explanations to the main research outcomes. Implications and future recommendations could be elaborated. 

Response. Thank you. We have added this part and also the Implications and future recommendations.

These findings support the authors’ hypothesis that being victim may lead to psychological trauma, and the individual resilience might play role in helping a victim back on track again[43]. However, another explanation concerning post-traumatic growth that might not be sufficiently addressed. For example, psychological trauma due to bullies may persist and interfere with the growth of a sense of self even though they receive some intervention or training in social skills [44].

Implications and future recommendations

The finding suggests the importance of identifying victim conditions when social skills and self-esteem associations were not observed as expected among school adolescents. Because self-esteem is a link to other mental health issue, especially depression[56], a screening for bullying-victim should be prioritized. Future research may focus on identifying the variable that would moderate the victim status, e.g., perceived social support[57,58].

Reviewer 2 Report

Major revisions are required

Please add research questions and address them in discussion section.

Please revise the conclusion section

Author Response

Reviewer 2

1.check phrasing (line 46)

Response. We have revised the whole paragraph. It’s now read.

The school system in Thailand has developed locally with a substantial focus on Thai culture and values. Schools with a Western education model were just infused in the early 19th century. Part of that is because Thailand is one of the few countries in the world that Western powers have never colonized. In recent years, international education in international schools has been widespread and has risen rapidly[7]. However, it is deemed for expatriates and high-income families, whereas a public-school education in Thailand until Grade 9 is free of charge. In addition, the students in international schools come from different ethnic and cultural backgrounds. Although low socioeconomic status may be significantly associated with bullying, it does not mean that bullying never exists among adolescents with high socioeconomic status, as in international schools[8-10]. However, reports regarding bullying in international schools, either in Thai or abroad, remain limited.

  1. ok, it’s good lit review (line 62)

Response: Thank you.

  1. maybe, more elaboration is needed on this part (line 102)

whereas another study showed that time perspective dimensions (feel- 102 ings, frequency and relation) moderated the associations between traditional and cyber- 103 bullying victimization and self-esteem among adolescents[23].

Response. For more understandable, we revised the statements as follows.

In contrast, another study showed the reciprocal relationships between adolescents' school victimization and self-esteem by that both variables predicted each other in the different time frames. In contrast, teacher support was the mediator between these two variables[24].

  1. maybe, more elaboration is needed on research question (line 126)

Response. We have revised this part. It’s now reads.

Even though both social skills and victimization have effects on self-esteem; however, it remains unclear whether the relationship between social skills and self-esteem depends on the condition of being bullied or not. To put other words, whether it has the interaction (or moderation) effect between social skills and victimization on self-esteem. More importantly, such a relationship in a setting like an international school, where the environment and socioeconomic status is unique and different from public school in Thailand, remain unknown.

As it is already known, the effect of bullying victimization on psychological distress is enormous, to the extent that some might experience suicide. In addition, low self-esteem, fear, distress, and depression can commonly occur. We propose that among adolescents who have never been bullied, social skills might directly affect the self-esteem level as acquiring more skills should provide individuals confidence and a sense of accomplishment. On the contrary, those experiencing being bullied might not gain the same benefit as the nonvictim because their sense of self-worth might be compromised. Therefore, the relationship between social skills and self-esteem in the latter group might not have a significant effect. This study aimed to explore the moderation effect of victim condition and social skills on self-esteem. The authors hypothesized that the moderating effect of victimization on the relationship between social skills and self-esteem would exist among the se adolescents. The authors' other hypotheses included the positive correlation between social capital and self-esteem and a negative correlation between discrimination and self-esteem.

The authors hypothesized that the magnitude of the relationship between 126 social skills and self-esteem in the victim condition was less robust than in the nonvictim 127 condition

  1. Please explian the design in detail (line 130)

This study employed a cross-sectional and correlational design conducted among students aged 13 to 18 in international schools in Thailand from 2021 to 2022.

6.Please provide more detail regarding participants (line 132)

Response. The participants were male or female students, aged 13 to 18 attending international schools in Thailand. Inclusion criteria comprised students who could understand, read and write English fluently. The exclusion criterion consisted of students with visual impairment.

7.Only with visual impairment – on other type of impairment (line 135)

Response. Yes. Only visual impairment because those with visual impairment cannot read the document well using a standard tool like other students. Those with hearing impairment were not excluded as long as they understood the language.

  1. provide more detail (132)

Response. Thank you. We have revised as follows.

The correlation between bullying victims and self-esteem was approximately 0.3[33]based on a meta-analysis of 936 studies investigating the relationship between self-esteem and bullying behaviour (i.e., perpetration and peer victimization).

  1. please address this limitation in this research(line 139)

Response. Thank you. We have added this in the limitation.

This study encountered limitations, First, the finding cannot represent the international school student due to the small number of subjects and the non-random sampling method

10.ok, did this affect the results (line 150)

Response. A compensation of 100 THB (approximately 3 USD) was small amount for the participants who are from the high-income family otherwise they cannot be admitted to the international schools due to the high cost of tuition fee. This amount then should not affect the motivation for the students to participate.

  1. please add appendix (line 153)

Response. We have  added the questionnaire in the supplementary files.

In addition to the questionnaires regarding sociodemographic data, the following measurements were provided for the participants (see the supplement files).

  1. please explain (line 156)

Response. We have revised as follows.

This tool looks at seven types of bullying, including bullying, verbal abuse, ostracizing (expelling from a group)

  1. please elaborate (line 160)

Response. We have revised for more clarification as follows.

Researchers have found good internal consistency of the OBVQ (Cronbach’s alpha ranged from 0.8 to 0.9)

  1. you need to explain the statistical analysis (line 166)

Response. We have revised for more clarification as follows.

The internal consistency of the RSES was good (Cronbach's alpha was 0.86)

  1. ok (line 182)

Thank you

  1. ok (line 205)

Thank you

  1. ok (line 211)

Thank you

  1. please provide more elaborated explanation (line 214)

Response. We have revised as follows.

Most of the sample were in grade 12 and attended day school. Over a half reported that their academic performance was good to excellent. Over 90% reported no physical or mental illness, but about 20 % experienced alcohol abuse. Most of the participants had never been bullied (84.3%).

  1. ok (line 221)

Thank you

  1. please explain (line 223)

Response

We have revised as follows.

Table 3 shows the mean and standard deviation of each variable. Victim status was positively related to self-esteem scores (p <.01). In contrast, self-esteem scores were positively related to social capital and social skills but negatively related to discrimination index scores (p<.01).

  1. please explain the table (line 233)

Response. We have revised as follows.

This interaction signifies that the social scores and self-esteem relationship depend on the existence of victim status. For example, model 3 indicated the variances of self-esteem increased from 23.8% to 33.2% when the interaction term was added to the model, denoting the effect victim status has on this relationship.

  1. ok, more interpretation of the results is needed, please address each research question (line 261)

Response

We have broken it down to three points, the main, the second and the third hypothesis.

  1. more elaboration regarding the discussion is needed—please address each research question in a structured way (line 284)

We have discussed based on these three points ; the main, the second and the third hypothesis.

Racial discrimination should be a new paragraphe.

That’s correct. Thank you.

  1. this should be part of the conclusion (line 289)

Response.

This part is concerning strength and limitation of the study, which are necessary. The authors thought that it should be kept separated from the other parts.

  1. please revise this section—you need to have a proper conclusion section (line 299)

Response. We have revised it as follows.

This study demonstrated the moderation effect of victim condition on the relationship between social skills and self-esteem, highlighting the importance of identifying victim conditions when social skills and self-esteem associations were not observed as expected among school adolescents. The association between social capital and discrimination and self-esteem were supported among international school adolescents. However, a longitudinal study to confirm causal relationships should be encouraged. Further research on providing appropriate interventions and social skills training in the victim group is warranted. 

  1. please add recent references 2020-2022 (line 322)

Response.

We have replaced by Mohan et al.

Mohan, T.; Abu Bakar, A.Y. A systematic literature review on the effects of bullying at school. SCHOULID: Indonesian Journal of School Counseling 2021, 6, 35, doi:10.23916/08747011.

Reviewer 3 Report

Thank you very much for giving me the opportunity to review this manuscript. The idea of your article is interesting, my recommendations are the following:

Abstract

It would be recommended to specify how "common" it is, it would be more specific "a major problem" as in the introduction.

It would be recommended to include the sex variable due it is an important variable. We don´t know it there are 102 girls, boys…

It would be recommended to include the mean and Sd in the abstract too.

Introduction

It would be recommended to include differences between the sexes in this type of victimization because it is an important variable

it would be recommended to separate ex and sex orientation (not “sex or sexual orientation”). There are no interchangeable variables such as “race or ethnicity” (line 54).

Methods

In participants

it would be recommended to put results in its specific part.

If in line 136 the systematic review is based on another paper, it is needed to put it.

Results

it would be recommended to develop the tables, not only put it.

In this part appears some information as sex or mean and sd of the age that it is not named before. It would be recommended to give it coherence.

Discussion

It would be advisable to show if there are differences by sex here as well. This disaggregation, which is so important in victimization, it is not found in the majority of the document.

Limitations

It would be recommended to show the possibility that children with low self-esteem can also be potential victims, especially if they lack the social skills to defend themselves.

Author Response

Reviewer #3

  1. It would be recommended to specify how "common" it is, it would be more specific "a major problem" as in the introduction.

Response. We have revised as follows.

Bullying is a major problem and can be found in schools regardless of being public or private, elementary or secondary, urban or rural

  1. It would be recommended to include the sex variable due it is an important variable. We don´t know it there are 102 girls, boys…

Response. We have revised as follows.

A total of 102 students participated, 63% female. The mean age of the participants was 16.57(SD = 1.42). The victim was 16(15.7%), the mean (SD) for SC, SS, ADDI, and RSES was 7.82(2.37), 44.45(9.40), 12.33(9.82), and 27.85(5.31), respectively.

  1. It would be recommended to include the mean and Sd in the abstract too.

Response.

Results. A total of 102 students participated, 63% female. The mean age of the participants was 16.57(SD = 1.42). The victim was 16(15.7%), the mean (SD) for SC, SS, ADDI, and RSES was 7.82(2.37), 44.45(9.40), 12.33(9.82), and 27.85(5.31), respectively. 

Introduction

  1. It would be recommended to include differences between the sexes in this type of victimization because it is an important variable

Response. We have added the results regarding sexes

Regarding the relationship between sex and bullying, 37 males (83.8%) and 65 females (90.8%) had never bullied others; 12 participants were involved with bullying. The Pearson chi-square test indicated no significant relationship between sex and frequency of bullying, Pearson's χ²(1) = 1.108, p=.292. In Victim experience, 31 males (83.8%) and 55 females (84.6%) had never been bullied; 16 participants reported being victims. The Pearson chi-square test indicated no significant relationship between sex and frequency of being a victim, Pearson's χ² (1) = 0.012, p=.912. In the bystander group, 21 males (56.8%) and 38 females (58.5%) had never been bystanders. Forty-three participants reported being bystanders. The Pearson chi-square test indicated no significant relationship between sex and frequency of being a bystander, Pearson's χ²(1) = 0.028, p=.867.

  1. it would be recommended to separate ex and sex orientation (not “sex or sexual orientation”). There are no interchangeable variables such as “race or ethnicity” (line 54).

Response. Thank you. We have revised that.

Methods

In participants

it would be recommended to put results in its specific part.

Response. Thank you. We have checked and removed number of sample size, and proportion of sex from Method to the results part.

If in line 136 the systematic review is based on another paper, it is needed to put it.

Response. We have revised this part based on the other reviewer’s comment. The information is needed to calculated sample size.

Results

it would be recommended to develop the tables, not only put it.

Response. I am not sure I understand this comment correctly. I think the reviewer wants us to provide leading statements of the result before mentioning Table 1. We have added this part as follows.

Among 102 participants, 63.7% were women. The average age of the participants was 16.55 years (SD = 1.39). Most of the sample were in grade 12 and attended day school. Over a half reported that their academic performance was good to excellent. Over 90% reported no physical or mental illness, but about 20 % experienced alcohol abuse. Most of the participants had never been bullied (84.3%).

In this part appears some information as sex or mean and sd of the age that it is not named before. It would be recommended to give it coherence.

Response. Please see above.

Discussion

It would be advisable to show if there are differences by sex here as well. This disaggregation, which is so important in victimization, it is not found in the majority of the document.

Response. Thank you for you suggestion. We have analyzed sex and bullying and added these results.

Limitations

It would be recommended to show the possibility that children with low self-esteem can also be potential victims, especially if they lack the social skills to defend themselves.

Response. Thank you for this suggestion. We have added this point in discussion part as follows.

Conversely, adolescents with low self-esteem can be potential victims because they lack the social skills to defend themselves.

Round 2

Reviewer 1 Report

Key critical points are still the sample size and the originality, i feel there are a number of interesting points regarding the context of Thailand, and there are good improvements.

Introduction: I would add a little bit more about the Thai culture and what makes it any different from western cultures (Check cultural values) where many of such studies took place. Although the authors mentioned International schools, what is it about these schools that makes "bullying" worthy of research? is it the assumption that international schools are abiding by better standards? better teachers? and hence better behaved students? or is it that intentional schools might lead to different experiences of bullying. I think think this part could still be developed beyond surface level. 

Methods: Generally clear, please avoid describing "females" as "women" and keep it consistent.

Results: generally good, but need to attend to punctuation. 

Discussion: Still not fully developed, i find it difficult to find an answer to the "so what question?" it is really important to stress the importance of your study, and the moment you are not doing it justice mainly in terms of "practical implications" "policy related implications" this can be developed better to guide readers. For example if i was a psychologist/counsellor/teacher, how could i benefit from the findings? 

Although sample size was mentioned as a limitation, it is important to talk about the challenge of recruiting from private schools. Cyberbullying need to be addressed in your recommendations for future research, and also comparing  private and public schools and while exploring the "nature of bullying" in Thailand. 

Author Response

Dear Editor and reviewers,

We are appreciated your valuable comments. Please see below our response to those comments.

1.Introduction: I would add a little bit more about the Thai culture and what makes it any different from western cultures (Check cultural values) where many of such studies took place. Although the authors mentioned International schools, what is it about these schools that makes "bullying" worthy of research? is it the assumption that international schools are abiding by better standards? better teachers? and hence better behaved students? or is it that intentional schools might lead to different experiences of bullying. I think think this part could still be developed beyond surface level. 

  1. Thank you for these comments. We have revised as suggested. It now reads,

The school system in Thailand has developed locally with a substantial focus on Thai culture and values. Thailand is a collectivistic culture[7], and Thais tend to be more altruistic and helpful than people in individualistic cultures, for which these characteristics are attributable to  Theravada Buddhism[8]. Seniority and social hierarchy may play a role concerning school environment in public schools in Thailand compared those in Western cultures[9]. However, no evidence has been reported regarding how culture impacts on bullying. 

In Thailand, schools using a Western education model were just infused in the early 19th century. Part of that was because Thailand is one of the few countries in the world that Western powers have never colonized. In recent years, international education in international schools has become widespread and has risen rapidly[10]. However, the system is deemed for expatriates and high-income families, whereas a public-school education in Thailand until Grade 9 is free of charge. It has been generally accepted that international schools in Thailand abide by international standards offering high quality campuses, modern facilities, a plethora of learning resources and a variety of extracurricular activities. Because of that, bullying is expected to be non-existent.

However, bullying is a multi-faceted issue, which is best understood in the societal context in which it occurs[11, 12]. One of those are different ethnics and cultural backgrounds. Even though such differences may be an advantageous side in learning, it can also constitute a downside as it is open to racial discrimination, leading to bullying[13]. Other factors include socialization - how well individuals socialize with others, the number of friends and students socializing with friends who bully. Taken altogether, that raises the question regarding bullying in this setting. Although few reports have been published of bullying in the international school, it has been hypothesized that bullying might exist.

2.Methods: Generally clear, please avoid describing "females" as "women" and keep it consistent.

2.Response.  We have corrected that. Thank you.

3.Results: generally good, but need to attend to punctuation. 

3.Response. We have checked all. Thank you.

4.Discussion: Still not fully developed, i find it difficult to find an answer to the "so what question?" it is really important to stress the importance of your study, and the moment you are not doing it justice mainly in terms of "practical implications" "policy related implications" this can be developed better to guide readers. For example if i was a psychologist/counsellor/teacher, how could i benefit from the findings? 

  1. 4. Thank you for pointing these out. We have revised as suggested. It now reads,

This study’s main hypothesis was to explore the bullying victims’ conditions regarding the relationship between social skills and self-esteem. The findings confirmed that the association between the level of social skills and the level of self-esteem depends on the bullying victims’ conditions. For example, the relationship was expectedly positive among adolescents who had never been bullied. In contrast, the relationship between social skills and self-esteem was insignificant among students who had been bullied. Bullying victims’ conditions appear to be the third variable influencing the relationship because moderating effects between bullying victims’ conditions and social skills are significant. These results highlight the effect of the victimization experience on self-esteem through, in this case, social skills. The psychological effect of the victimization experience may persist even though the penetrations have ended[11].  In fact, social skills are merely one example involved in bullying victimization and self-esteem. Other promoting behaviors or skills might be observed on self-esteem along the same line as social skills. Therefore, it would be essential not to allow bullying victimization to go “undetected”.

In general, social skills provide confidence for those having mastered them[46, 47]. However, research findings concerning the relationship between social skills and self-esteem among adolescents were seemingly inconsistent. One study revealed a significant effect of social skills training programs on raising self-esteem and decreasing physical aggression[48]. Another study showed that social skills training did not increase self-esteem among these male adolescents[49]. Notably, both studies did not obtain bullying data in their studies. The authors assumed that those adolescents in the latter study might have been bullied because a positive association between social skills training and self-esteem was not illustrated as expected. Many skills-based bullying interventions aim to improve social skills among victims of bullying; that will not stop bullying from happening but could improve resilience. However, if the trauma from bullying is deep, it may not be recovered easily and may require additional and special intervention.  

The authors hypothesized that being a bullying victim may lead to psychological trauma, and individual resilience might play a role in helping a bullying victim back on track[50]. Another explanation concerns posttraumatic growth that might not be sufficiently achieved. For example, psychological trauma due to bullies may persist and interfere with the growth of a sense of self even though they receive some intervention or training in social skills [51].

Interestingly, some investigators found contrasting results. Students with low self-esteem and lack of confidence in their social and communication skills were more likely to be victims of bullying[52]. In such cases, self-esteem tends to mediate the relationship between the social skills and being a bullying victim. Quite possibly, a situation could occur either way or even involve a vicious cycle. Conversely, adolescents with low self-esteem can be potential victims because they lack the social skills to defend themselves. However, the present study suggests that identifying the bullying victims’ status is crucial, so that additional or special interventions may be provided along with social skills training. It would be likely that these adolescents, prone to be bullying victims, might experience aforehand psychological  trauma or personality problems, requiring recovery as posttraumatic growth before acquiring further new skills[51].

The second hypothesis confirmed the existing relationship between social capital and self-esteem. Consistent with the related studies, social capital, defined by the number of friends, trusting friends and the extent to which an individual feels autonomy and independence, did predict the level of self-esteem, but was unrelated to the victim or nonvictim status. The related evidence has shown that such results are inconsistent [53, 54]. The authors ascribed these findings to the relatively high level socioeconomic status and low prevalence of bullying among the participants in such types of schools.

The third hypothesis confirmed the existing relationship between discrimination and self-esteem. In line with many studies, discrimination was found to be a significant negative predictor of self-esteem[55]. Researchers have also found the association of racial discrimination and other outcomes among children and adolescents, e.g., well-being or health[56, 57], and as a risk factor for depressive symptoms and substance abuse[58]. This study explicitly noted racial discrimination in an international school. Even though racial discrimination was unrelated to victimization, it remained significantly associated with self-esteem. Further research regarding strategies to reduce the strength of the relationship between discrimination and low self-esteem should be warranted.

Implications and future recommendations

The finding suggests the importance of identifying bullying target’s conditions on self-esteem through social skills, and self-esteem associations were not observed as expected among school adolescents. Because self-esteem constitutes a link to other mental health issues, especially depression[59], a screening for bullying targets should be prioritized. The matters of victimization experience are essential and can be underdetected. Implementing screening programs can be carried out using a variety of methods and levels. For example, teachers or school counsellors can adopt a tactful approach to elicit such information from students, especially those with low self-esteem or who seem not to benefit from self-promoting activities. In contrast, school administrators may proactively devise an intermittent friendly screening tool to derive the student’s victimization experience as well as racial discrimination without questioning its existence in such a setting. Individual characteristics of students, family members, teachers and schools need to be targeted in bully prevention programs to reduce bullying and victimization in schools. Future research may focus on identifying variables moderating the bullying target’s status, e.g., perceived social support[60, 61]. Also recommended is exploring the issue of cyberbullying and comparing private and public-school experiences while investigating the "nature of bullying" in Thailand.

5.Although sample size was mentioned as a limitation, it is important to talk about the challenge of recruiting from private schools. Cyberbullying need to be addressed in your recommendations for future research, and also comparing private and public schools and while exploring the "nature of bullying" in Thailand. 

5.Thank you for this suggestion. We have added this point. It now reads,

Strength and limitations

Although many studies have been conducted regarding the association between social skills, bullying targets’ conditions and self-esteem, this study comprised one of the first to demonstrate the interaction effect of being a bullying target and social skills concerning their level of self-esteem. This positively contributed to our attempts to break through the bullying target’s social skills and self-esteem; thereby, intervening and reducing the incidence and adverse effects of bullying. This study encountered limitations. First, the finding cannot represent the international school student due to the small number of subjects and the nonrandom sampling method. Second, as almost all international schools are private and for-profit organizations, reputation became an issue; therefore, for the researcher to recruit participants was challenging. Third, the varying school factors were not accounted for in the analysis, and finally, the cross-sectional design could not conclude or infer a causal relationship.

Reviewer 2

Comments and Suggestions for Authors

The conclusion section should be revised--it should be more detailed and show the significance of the study, its imitations and implications for future studies

Response

We have revised as follows.

  1. Conclusion

This study demonstrated the significant moderation effect of bullying victim’s conditions on the relationship between social skills and self-esteem, denoting that the relationship between social skills and self-esteem depended on the status of bullying victimization. The study practically highlighted the importance of identifying bullying victim’s conditions when social skills and self-esteem associations among school adolescents were not observed as expected. As anticipated, a significant relationship was observed between social capital, discrimination, and self-esteem. However, the small sample size and convenience sample prevent it from constituting a representative sample of international students in Thailand. In addition, the cross-sectional design limits determining a causal relationship among variables. Future studies with a more considerable sample size with random sampling should be encouraged. A longitudinal study is required to confirm a causal relationship. Also recommended is exploring the issue of cyberbullying and comparing private and public-school experiences. Moreover, research on providing appropriate interventions and social skills training in the bullying victim group is warranted. 

Reviewer 2 Report

The conclusion section should be revised--it should be more detailed and show the significance of the study, its imitations and implications for future studies

Author Response

Dear Editor and reviewers,

We are appreciated your valuable comments. Please see below our response to those comments.

Reviewer 2

Comments and Suggestions for Authors

The conclusion section should be revised--it should be more detailed and show the significance of the study, its imitations and implications for future studies

Response

We have revised as follows.

  1. Conclusion

This study demonstrated the significant moderation effect of bullying victim’s conditions on the relationship between social skills and self-esteem, denoting that the relationship between social skills and self-esteem depended on the status of bullying victimization. The study practically highlighted the importance of identifying bullying victim’s conditions when social skills and self-esteem associations among school adolescents were not observed as expected. As anticipated, a significant relationship was observed between social capital, discrimination, and self-esteem. However, the small sample size and convenience sample prevent it from constituting a representative sample of international students in Thailand. In addition, the cross-sectional design limits determining a causal relationship among variables. Future studies with a more considerable sample size with random sampling should be encouraged. A longitudinal study is required to confirm a causal relationship. Also recommended is exploring the issue of cyberbullying and comparing private and public-school experiences. Moreover, research on providing appropriate interventions and social skills training in the bullying victim group is warranted. 
